# Improving tuberculosis diagnosis in South Africa's private sector: The results of a pilot public-private mix intervention in eThekwini health district

Jody Boffa [1,2*], Tsholofelo Mhlaba[3*], Buyisile Chibi[4], Mergan Naidoo [5,6], Keeren Lutchminarain[7,8], Khine Swe Swe-Han[8,9], Jeremiah Chikovore[4], William Mapham [10], Sizulu Moyo[4,11]

1 The Aurum Institute, Johannesburg, South Africa, 2 Centre for Rural Health, University of KwaZulu-Natal, eThekwini, South Africa, 3 Discipline of Public Health Medicine, University of KwaZulu-Natal, eThekwini, South Africa, 4 Public Health, Societies and Belonging Division, Human Sciences Research Council, uMgungundlovu, eThekwini, and Cape Town, South Africa, 5 Discipline of Family Medicine, University of KwaZulu-Natal, eThekwini, South Africa, 6 Wentworth Hospital, eThekwini, South Africa, 7 National Institute for Communicable Diseases, Johannesburg, South Africa, 8 Medical Microbiology Department, University of KwaZulu-Natal, eThekwini, South Africa, 9 National Health Laboratory Service, eThekwini, South Africa, 10 Vula Mobile, Cape Town, South Africa, 11 School of Public Health, University of Cape Town, Cape Town, South Africa

* jboffa@tbthinktank.org (JB); mhlaba@ukzn.ac.za (TM)

**Data availability statement:** All data relevant to this paper are publicly available through The Aurum Institute's data repository. Requests can be directed to lmchunu@auruminstitute.org.

**Funding:** This study was funded by the Stop TB Partnership's TB REACH programme in the form of a grant [W8_ZAF_PPE-UKZN to JB and

## Abstract

While tuberculosis (TB) is primarily addressed in South Africa's public sector, people with TB also present to private sector General Practitioners (GPs), where TB may be missed or treatment delayed. We introduced a pilot project in a high-TB burden health district to connect private GPs to free public sector TB testing. We aimed to gauge GPs' willingness to participate and describe TB patterns in the private sector. GPs practicing in metropolitan eThekwini from May 2021-March 2022 were invited to participate. Recruited GPs were provided sputum specimen jars, specimen transportation, and access to free TB testing through the National Health Laboratory Service for clients with TB-like symptoms. A customised electronic form on an established medical referral application (Vula) was developed to record client information, initiate specimen transport, share real-time test results, and communicate management guidance. Of the 313 eligible GPs, 158 (50.5%) agreed to participate, among whom 61 (38.6%) submitted at least one client specimen (median=6, IQR=2-12). Specimen yield (17.6%) and quality (99.7%) were high. One-hundred-seven clients were diagnosed with TB, 39.3% were female and 48.6% were living with HIV. Three clients (2.9%) were diagnosed with drug-resistant TB. One hundred people with TB (93.4%) were linked to treatment, 96.0% in the public sector, in an average of two days (IQR 1-5), with 88/100 completing treatment in a median 182 days (IQR=170-194). Two people with TB died before diagnosis by culture and six died during treatment, resulting in 7.5% case fatality (8/107). User-prompting to check HIV status significantly improved the frequency with which GPs enquired about HIV compared to a previous study (88.4% versus 25.7%, p<0.0001). One-fifth (19.5%) of GPs submitted specimens without

TM]. The Bill and Melinda Gates Foundation in South Africa supported the cost of reflex testing [to JB and TM]. Stop TB Partnership was involved in the initial study design to ensure it aligned with the purpose of their TB REACH grant programme which aims to implement innovative and sustainable TB-related interventions in high-burden settings. Funders had no role in data collection and analysis, decision to publish, or preparation of the manuscript.

**Competing interests:** The authors have declared that no competing interests exist.

monetary incentives and helped link 100 clients to TB treatment expeditiously, suggesting a successful pilot and a workable model for improving TB management in South Africa's private sector.

## Introduction

Tuberculosis (TB) remains among the top causes of death by an infectious disease globally [1]. South Africa is among the 30 highest TB burden countries globally and one of 10 reporting high burdens of TB, HIV-related TB, and drug-resistant TB simultaneously [1]. While TB is primarily addressed in South Africa's public sector, recent evidence suggests TB may be missed or delayed in the private sector. In a 2019 study in Cape Town and eThekwini, mystery clients presenting to private General Practitioners (GPs) with textbook TB-like symptoms were sent away without a TB test or onward referral in 57.4% of interactions recorded with GPs, often with broad-spectrum antibiotics dispensed or prescribed [2]. No GPs referred clients directly for TB testing by GeneXpert, which is the first-line test according to South Africa's national TB guidelines, because of the cost of testing in the private sector [2–4]. Enquiry about HIV status in this high prevalence setting was also extremely low at 25.7% [2].

In South Africa, primary healthcare services in the public sector are primarily led by clinical nurse practitioners and serve approximately 80% of the population [5]. TB management has also been evaluated in the public sector using the mystery client model. In 84% of healthcare visits, clinical nurse practitioners initiated sputum specimen collection for testing by GeneXpert (now Ultra), which is freely available in the public sector, and HIV tests were offered in 47% of visits [6].

In the private sector, GPs are typically the first point of contact and cater to both the 15.8% of the population with health insurance [7] and those who are willing to pay cash – typically between US$10-30 per visit. Reasons for seeking care in the private sector for those without medical insurance may relate to shorter wait times, convenient location, and perceived quality of care [8]. Although TB disproportionately affects the underprivileged, its high prevalence in South Africa (468 per 100,000 population [1]) means that people with TB-like symptoms do not exclusively present to the public health sector.

In an effort to understand the burden in private practice and reduce potential gaps in the TB care cascade, we developed an intervention that connected private GPs to free GeneXpert Ultra tests through the public sector using an existing mobile medical referral application called Vula to initiate specimen transport. Our objectives were to i) describe the uptake of the intervention by GPs in this high-TB burden setting, ii) describe patterns of TB diagnosed in the private sector, and iii) test our pilot intervention for its ability to improve enquiry into HIV status for people presenting with TB-like symptoms.

## Methods

### Design

GPs typically refer to the public sector for TB testing and management [2]. We therefore utilised a single-arm experimental design to pilot the study, as there were no pre-intervention data on TB diagnoses and treatment among private GPs. At the project outset, consenting GPs were asked to attend two one-hour virtual training sessions on sputum specimen collection and use of Vula, an established medical referral application with a customised electronic form for the study (Fig 1). The electronic form was developed to record client information, initiate specimen transport, share real-time test results with clients via text message, and communicate management guidance to GPs. Trainings were offered on evenings and weekends to

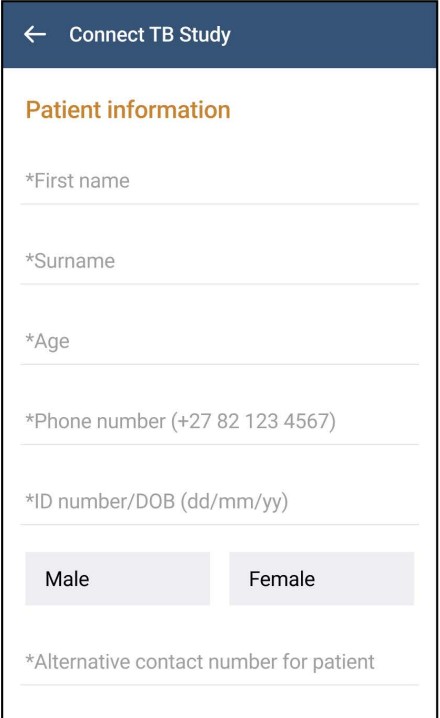
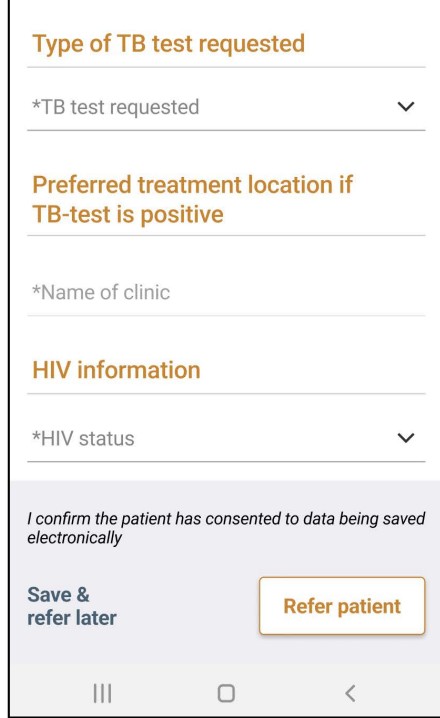
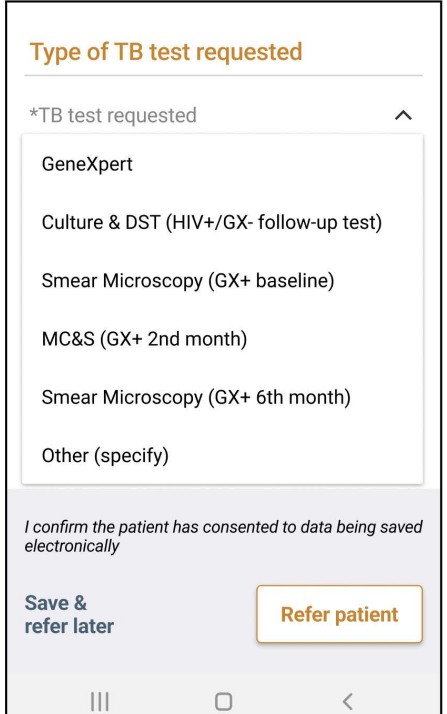
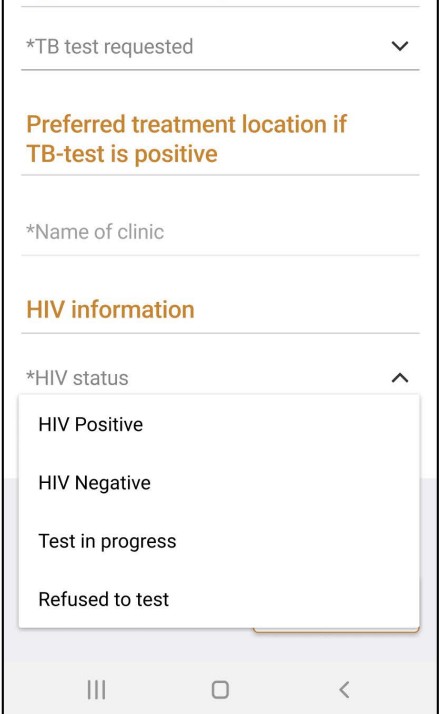

**Fig 1. Vula electronic form completed for private sector clients with TB-like symptoms.**

accommodate GPs' schedules and recordings were made available for those who could not attend. Training time was not financially compensated, but participating GPs were awarded Continuing Professional Development points. As training sessions became a barrier to active GP participation, we soon developed short-form training materials and employed research assistants to provide on-the-spot training in Vula and specimen collection at the time of GP consent.

GPs were provided with specimen jars and labelling bags that clients would use to collect sputum in a private, ventilated area of the clinic or immediately outside the clinic building. The sputum specimens were tested using GeneXpert Ultra by the National Health Laboratory Service (NHLS) in collaboration with the KwaZulu-Natal Department of Health and eThekwini District Health Office at no cost to the GP or client. Transportation of specimens from the GP's offices to the testing lab was provided throughout the project by study-employed drivers. Drivers were trained to complete NHLS requisition forms based on information provided in Vula and delivered specimens to a central NHLS laboratory where a project technician was employed to receive and test specimens.

Clients testing positive for TB also received telephonic adherence support through project-supported adherence facilitators. Findings on telephonic adherence support are reported elsewhere [9].

The intervention ran from 01 May 2021 to 31 March 2022.

## Participants

The offices of all private GPs registered with Government Employee Medical Scheme (GEMS), a large medical insurer, that fell within the bounds of the study area in eThekwini Municipality were approached in person by study staff to seek consent of the practicing GPs between 01 March and 20 December 2021. Site boundaries for recruitment extended to Amanzimtoti in the South, Pinetown and KwaNdengezi to the East, and Tongaat to the North (see Fig 2). GPs from practices with ≥ 95% of clients who paid by medical insurance were excluded from recruitment efforts, as we considered this a good proxy for higher socio-economic status of clientele and a lower proportion of clients with TB. Study staff made up to five attempts to reach GPs for consenting purposes before declaring a GP unreachable. Given its labour-intensive nature, GP recruitment was ongoing throughout the study.

## Data collection

GP demographics were collected at the time of consent. Sputum specimen requests initiated by GPs through the Vula app and specimen results from NHLS were recorded systematically in a password-protected project database along with client demographics. A separate password-protected database was maintained for clients diagnosed with TB, including treatment collection dates and client outcomes ascertained from adherence facilitators and verified against facility-level data systems (when clients were treated in the public sector) or the GP (when clients were managed in the private sector). Client HIV status was entered into the electronic form in the Vula application by the attending GP as one of: positive, negative, test in progress, or client declined to test.

## Analysis

We report study uptake, GP and participant characteristics, TB diagnoses, diagnostic yield, and specimen quality using frequencies. A TB diagnosis was defined as a positive test by GeneXpert Ultra or culture when follow-up test was indicated by national guidelines, i.e. in the instance of people living with HIV with a GeneXpert-negative result, anyone with a

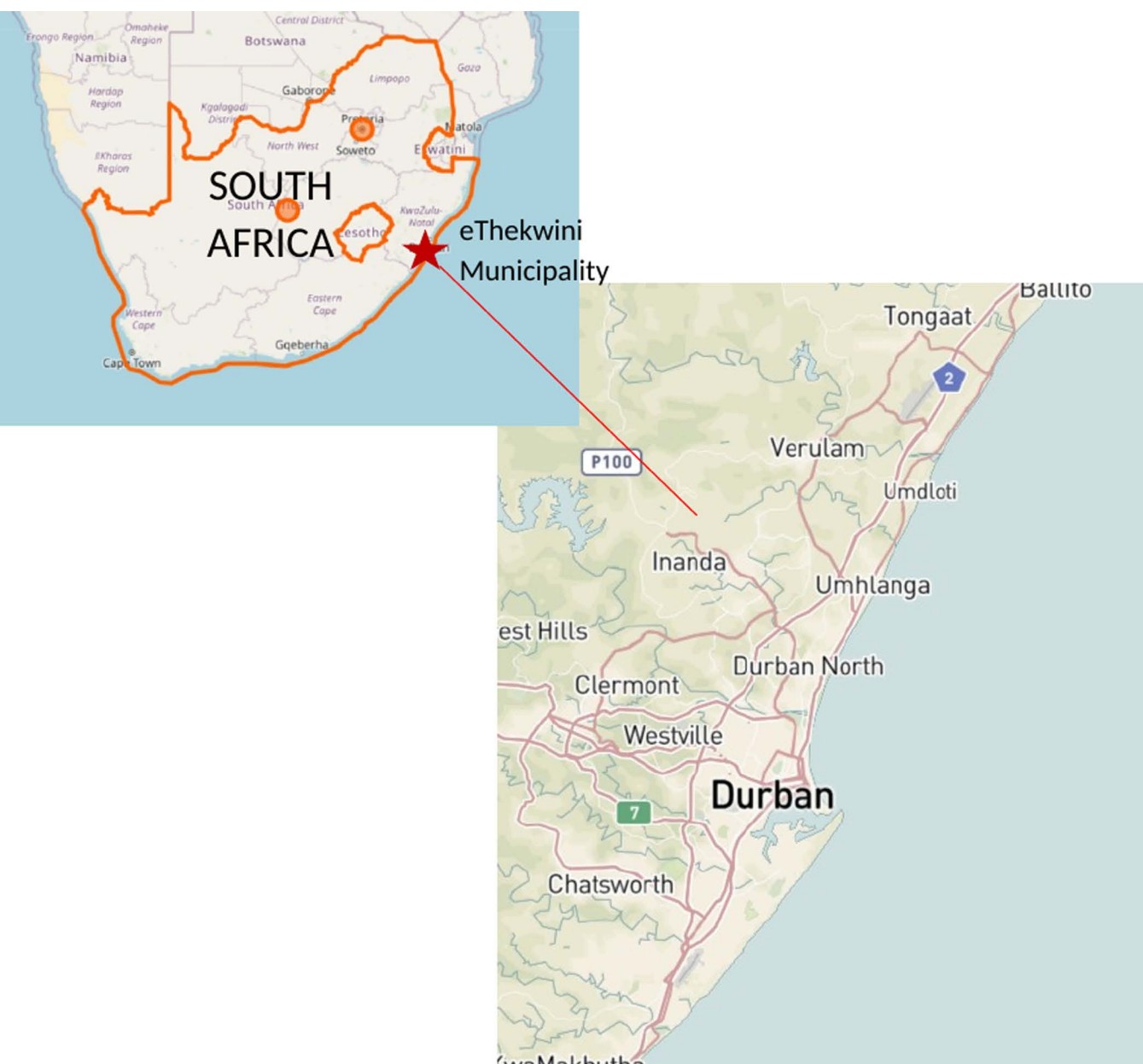

**Fig 2. Map of study site boundaries** [ 10,11]. *Source*: Maps from OpenStreetMap and OpenStreetMap Foundation, United States Geological Survey.

"trace" result on GeneXpert Ultra, or anyone with persistent symptoms one week after a negative GeneXpert Ultra result [3]. We defined drug-resistant TB as rifampicin resistance on GeneXpert Ultra testing or isoniazid and rifampicin resistance on reflex testing [3]. We used two-sample proportion tests in Stata 15 with p-values when comparing two proportions. We used two-way analysis of variance with p-values to compare means across multiple groups (e.g. physician characteristics by practice area).

We report time to treatment initiation as the difference in days between specimen collection and treatment start date recorded at public clinic or self-reported treatment start date for clients accessing TB treatment in the private sector. We report treatment duration as the

difference between treatment start date and the date of treatment completion noted in the TB notification database. Where treatment completion dates were not available (e.g. the client did not return to the clinic for smear microscopy after treatment completion), we report the difference between the start date of treatment and 30 days after the last date of medication collection as reported in treatment records or self-reported for those who purchased treatment through private pharmacies. As TB diagnosis and management data is not routinely collected in the private sector, and District line list data for the public sector are restricted, we were unable to make direct comparisons to general public or private sector averages in eThekwini during the same period. Instead, we compared client characteristics (sex and HIV status), drug resistance, treatment completion rates, and mortality numbers to national notification data for the most recently available years.

### Ethics

The study received ethical approval from the University of KwaZulu-Natal's Biomedical Research Ethics Committee (protocol reference number: BREC/00002279/2021) and administrative approvals from the Health Research and Knowledge Management Unit of the KwaZulu-Natal Department of Health (ref: KZ_202102_002) and the eThekwini District of Health. Written, informed consent was sought and obtained from all GPs who participated in the pilot. GPs asked clients to orally consent for their information to be shared with researchers for the pilot before data were entered into the Vula application. All participants were assured of confidentiality, with only aggregate-level outcomes reported.

The Vula smartphone application is end-to-end encrypted and compliant with South Africa's Protection of Personal Information Act. Its use is restricted to medical professionals registered with the Health Practitioners Council of South Africa.

## Results

### GP participation

A total of 158 GPs consented to participate of 313 approached (50.5%), contributing a total of 1432 practice months to the study (median 10.5 months, IQR 6-11). Of those consenting, 128 (81.0%) availed themselves for training on Vula and specimen collection and 61 (38.6%) submitted at least one sputum specimen (median=6, interquartile range [IQR]=2-12). Fig 3 displays the flow of private GPs in the district and specimens submitted. Characteristics of consenting GPs are described by sputum-specimen submission status in Table 1.

GPs practicing in townships, cities, and suburbs did not differ significantly by patient load (p=0.83), years in practice (p=0.83) or percent of cash-paying clients (p=0.92), but did differ significantly on consultant fee (p=0.002). Township GPs charged on average ZAR349, while city and suburb GPs charged an average of ZAR393 and ZAR407, respectively. Consultant fees did not differ significantly by area between those who did and did not submit specimens.

### TB patterns in the private sector

A total of 597 sputum specimens were submitted for testing for 579 clients across 11 months. Specimens were split for reflex testing when indicated, and seven additional specimens were collected for culture when HIV-negative clients had persistent symptoms after testing negative by GeneXpert Ultra, in accordance with national guidelines [3]. Two specimens were declined due to poor quality. Some 104/590 (17.6%) specimens tested positive for TB by GeneXpert Ultra, three of which were rifampicin resistant (2.9%). Three additional clients were diagnosed

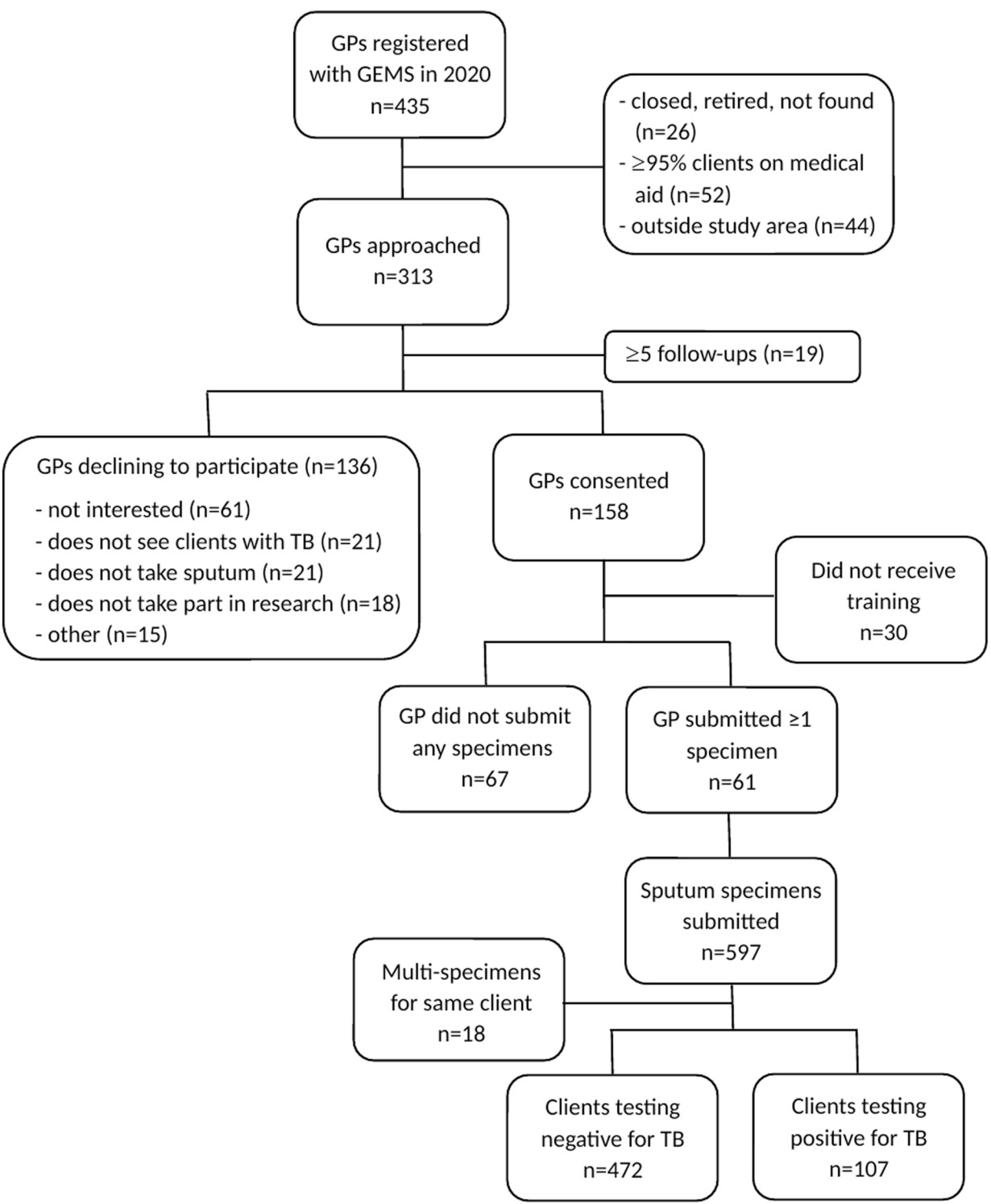

**Fig 3. Flow diagram of GPs and specimens submitted.**

**Table 1. GP Characteristics by sputum-specimen submission status.**

| Characteristic | No specimens | At least 1 specimen |
|---|---|---|
| Length of participation in pilot in months, median (IQR) | 10 (6-11) | 11 (10-11) |
| Sex, n (%) | | |
| F | 20 (58.8) | 14 (41.2) |
| M | 77 (72.1) | 47 (37.9) |
| Area, n (%) | | |
| City | 29 (50.9) | 28 (49.1) |
| Township | 55 (75.3) | 18 (24.7) |
| Suburb | 13 (46.4) | 15 (53.6) |
| Daily patient load, median (IQR) | 20 (10-35) | 20 (10-30) |
| Years in practice, median (IQR) | 23 (15-30) | 19 (11-27) |
| Percent cash-paying clients, median (IQR) | 40 (20-60) | 30 (15-50) |
| Consultant fee in ZAR, median (IQR) | 380 (300-430) | 380 (330-420) |

Abbrev: IQR=interquartile range, ZAR=South African Rand.

with drug-susceptible TB by culture. Two of these were diagnosed posthumously and all were HIV-positive. Table 2 compares TB yield and the proportion of unsuccessful specimens from the project to those tested in the public sector in eThekwini District over the same period.

Of the 579 clients for whom specimens were submitted, 278 (48.0%) were female and 196 (33.9%) were known to be living with HIV. A total of 65 clients indicated an initial preference to be treated in the private sector at the time of testing (11.2%). Overall, 96/100 (96.0%) of clients initiating TB treatment opted for treatment in the public sector due to access to free medication and follow-up care.

A total of 107 people were diagnosed with TB during the study, two posthumously. Three clients had drug-resistant TB: two females and one male, two of whom were living with HIV. Five clients with drug-susceptible TB died during treatment at an average of 56 days after treatment initiation (IQR 24-140), and one client with drug-resistant TB died during the ninth month of treatment. Among those alive at the time of diagnosis (n=105), 95.2% (n=100) were linked to treatment in a median of two days (IQR=1-5) and 83.8% (n=88) completed treatment in a median of 182 days (IQR 170-194). Three people experienced treatment failure; all of whom were restarted on treatment. Table 3 summarises client characteristics and outcomes.

Proportionally, our findings were similar to national TB notifications in terms of breakdown by sex (F=39.3% vs 40.9%, p=0.74), clients with known HIV status who are HIV positive (people living with HIV=52.0% vs 54.0%, p=0.69), drug-resistant TB (2.8% vs 3.5%, p=0.68) and treatment success (82.2% vs 79.0%, p=0.41) [1]. Case fatality in the study was 7.5% (8/107), significantly less than the estimated national average of 19.2% (p<0.01) [1].

**Table 2. Yield and quality of sputum specimens.**

| Sector | Xpert tests performed N | Unsuccessful specimens n (%) | p-value | Pulmonary TB detected n (%) | p-value |
|---|---|---|---|---|---|
| Private GPs | 590 | 2 (0.3) | | 104 (17.6) | |
| Public clinics | 183 434 | 3125 (1.7) | 0.01 | 14 538 (7.9) | <0.0001 |

**Table 3. Outcomes for clients diagnosed with TB, by sex, age and HIV status (N=107).**

| Demographic variable | Overall n (%) | Cured/ Completed treatment n (%) | initial loss to follow-up n (%) | Lost to follow-up n (%) | Died on treatment n (%) | Treatment failure n (%) |
|---|---|---|---|---|---|---|
| **Sex** | | | | | | |
| F | 42 (39) | 35 (40) | 2 (29) | 1 (33) | 3 (50) | 1 (33) |
| M | 65 (61) | 53 (60) | 5* (71) | 2 (67) | 3 (50) | 2 (67) |
| **Age group** | | | | | | |
| 15-34 | 39 (36) | 34 (39) | 1 (14) | 0 (0) | 1 (17) | 3 (100) |
| 35-49 | 42 (39) | 33 (38) | 3** (43) | 3 (100) | 3 (50) | 0 (0) |
| 50-64 | 23 (21) | 19 (22) | 2** (29) | 0 (0) | 2 (33) | 0 (0) |
| 65+ | 3 (3) | 2 (2) | 1 (14) | 0 (0) | 0 (0) | 0 (0) |
| **HIV status** | | | | | | |
| positive | 52 (49) | 40 (45) | 4* (57) | 3 (100) | 3 (50) | 2 (67) |
| negative | 48 (45) | 43 (49) | 1 (14) | 0 (0) | 3 (50) | 1 (33) |
| unknown | 7 (7) | 5 (6) | 2 (29) | 0 (0) | 0 (0) | 0 (0) |

*Includes two clients who died before diagnosis by culture.

**Includes one client who died before diagnosis by culture.

### Ascertaining HIV status

HIV status was entered at the time of specimen request for 528/597 samples (88.4%), including two instances in which GPs initiated serum-based HIV tests. Significantly more clients had data on HIV status than in a 2019 mystery TB client study of private sector GPs from the same population (p<0.0001, see Fig 4) [2].

## Discussion

In general, the pilot was successful. One half of eligible GPs agreed to participate (n= 158, 50.5%), 79.0% of whom underwent training (n=128). Of those trained, about half (n=61, 47.7%) submitted at least one sputum specimen (median=6, IQR2-12) over an average of 11 months (IQR=6-11). Specimen yield and quality were high at 17.6% and 99.7%, respectively. About half of specimens submitted came from men (52.0%) and a third from clients living with HIV (33.9%). One hundred seven clients were diagnosed with TB, more than half of whom were men (60.7%) and about half were living with HIV (48.6%). Three clients were diagnosed with drug-resistant TB (2.9%). One hundred people with TB were linked to treatment, nearly all in the public sector (96.0%) in an average of two days (IQR 1-5). Two people with TB died before diagnosis by culture and six died during treatment, resulting in a case fatality of 7.5% (8/107). User-prompting to check HIV status significantly improved the frequency with which GPs enquired about HIV compared to a previous study [2].

### GP participation

While GP uptake was lower than anticipated with 50.5% of eligible providers agreeing to participate, it was proportionally similar to the engagement of allopathic doctors (1654/3352 or 49.3%) in a pilot study aiming to increase TB testing and treatment in Mumbai, India between 2014-17 [12]. While the Mumbai study involved incentives for providers for positive tests and treatment completion [12], in the present study, 40.9% of eligible providers in a large urban setting of South Africa undertook training to collect sputum onsite and 19.5% submitted specimens (38.6% of engaged providers) over a period of 11 months, without monetary incentive.

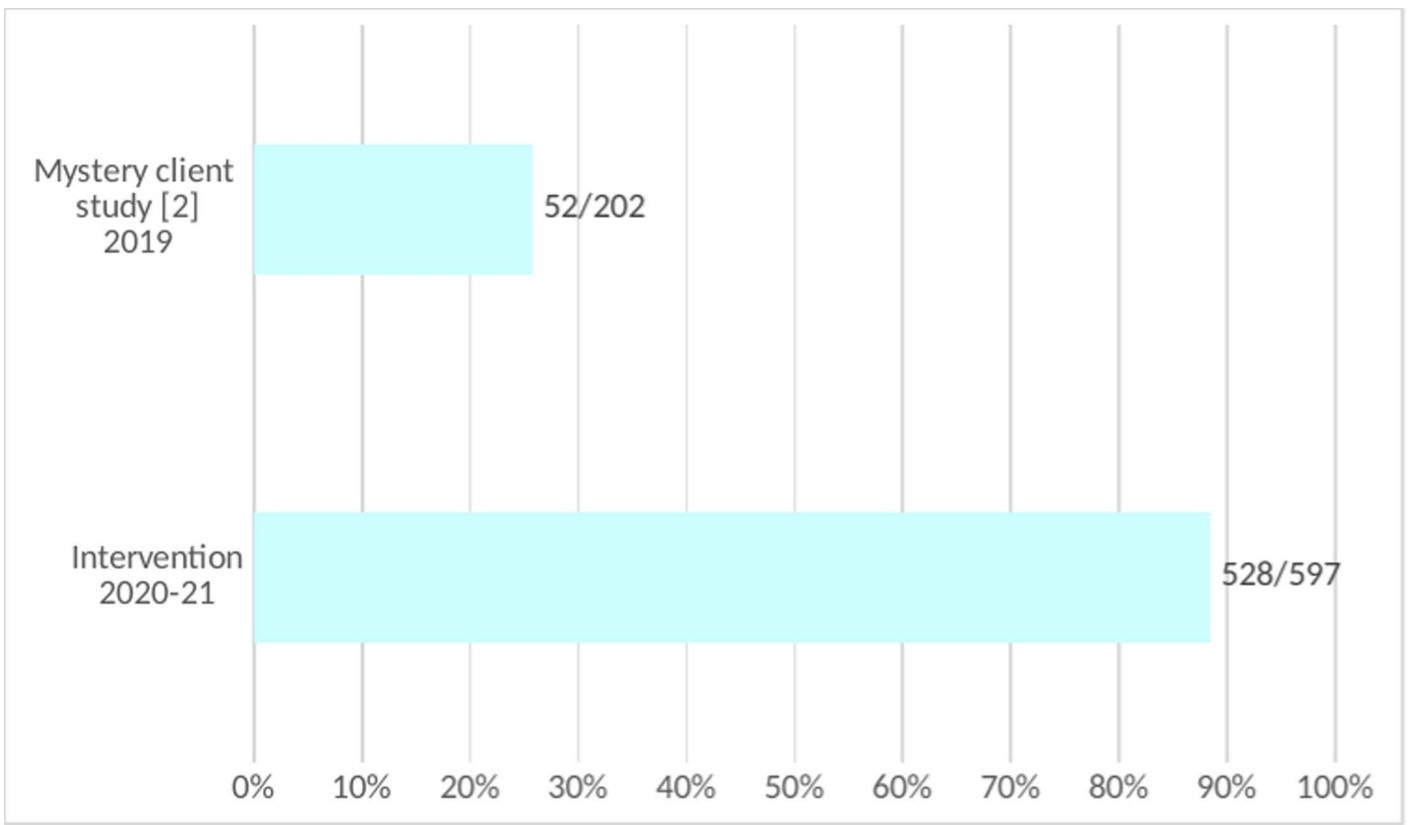

**Fig 4. Comparison of clients queried for HIV in present versus previous study.**

In the Mumbai study, a higher proportion of doctors submitted at least one specimen (62.8% vs 38.6% of engaged doctors) which may be explained by a combination of longer coverage time (3.5 years), availability of public stock for private doctors in Mumbai to treat clients with TB [12], and the present study overlapping with multiple COVID-19 outbreaks, leaving some doctors wary of initiating onsite sputum specimen collection.

About half of GPs recruited from practices in the city (28/57) and suburbs (15/28) submitted at least one specimen during the study. GPs practicing in townships represented the highest proportion of practitioners (46.2%), yet only a quarter of those submitting at least one specimen (18/61). Part of this difference may be explained by the lower average consultation fee charged by GPs in townships (RZAR349 vs ZAR382 and ZAR407 charged in the city and suburbs, respectively, p=0.002). Modest monetary incentives (e.g., ZAR100) per case identified may help to improve TB testing, especially in areas with lower consultation fees.

## TB patterns in the private sector

**Specimen yield and quality** Specimen yield and quality were high at 17.6% and 99.7%, respectively, suggesting that point of care specimen collection is possible in the private sector. GPs who took part in the study were trained on optimum sputum specimen collection techniques, the recency of which may explain improved quality. Other factors that may have resulted in better quality specimens in GP clinics may include more time to spend with each client to explain the process, faster time from collection to testing due to on-call transport, or clients with more severe symptoms being prioritised for testing compared to public clinics, making

successful sputum production more likely. The higher yield compared to eThekwini's public sector (7.9%) may be due to higher quality specimens, although it is also likely due to under-screening for TB, as one would expect a lower overall yield given the epidemiology of the setting. Between 2017 – April 2023, the national pulmonary TB yield by GeneXpert Ultra was 8.4% [13].

**Time to treatment initiation and completion** Among people initiating treatment (n=100), the median time to treatment initiation was two days from the time of GP visit (IQR=1-5), and 88.0% completed treatment. The median time to completion was 182 days (IQR 170-194). Although comparative data on diagnostic and treatment delays were not available – private sector data is not routinely collected, and public sector data was inaccessible – data from other studies suggests that treatment delays were markedly reduced through the intervention. In 2024, the average turnaround time for GeneXpert Ultra testing in South Africa's public sector was two days and clients are typically requested to return to clinic for sputum speci-men results after one week. A study of TB management in the private sector found that the majority of providers in eThekwini and Cape Town sent 57.4% clients with textbook TB-like symptoms away without TB test referral, while 38.6% referred clients to the public sector for work up, suggesting further delays in addition to the public sector average [2]. A 2011 study in a Cape Town township reported that 27% of enrolled participants with TB reported initially visiting a healthcare provider outside the public system for TB-like symptoms, with a median health system delay of 25 days (IQR 13–55) until treatment initiation compared to 13.5 days (IQR 7–31) in the public sector (p<0.01) [14]. Clients using non-public providers had a mean number of four visits (IQR 3–5) before treatment initiation, compared to three (IQR 2–4) in the public sector (p<0.001) [14].

**Client characteristics** Notwithstanding the high testing yield, the number of clients diagnosed in the private sector was low compared to the public sector. Clients diagnosed with TB were proportionally similar in sex and HIV status compared to national data (F=39.3% vs 40.9%, people living with HIV among status known=52.0% vs 54.0%, respectively), and a simi-lar proportion of people with drug-resistant TB was reported (2.8% vs 3.5%) [1]. Notably the male-female testing ratio was higher in the private sector (301/278 or 1.1:1) than was reported in a 10-year summary of public GeneXpert tests nationally (0.85:1) [13]. Much has been written on how men present to the South African public sector less often than their female counterparts, and generally at later stages of illness [15–17]. The present findings suggest that improving TB screening and test initiation in the private sector may be one way to reach more men. Case fatality was also lower in the study (7.5%) compared to national data (19.2%) [1]. This may be due in part to prompt diagnosis and treatment initiation, but is also likely attrib-utable to the routine telephonic support of adherence facilitators employed by the project [9].

## Ascertaining HIV status

Prompting private GPs to enquire about the HIV status of clients with TB-like symptoms significantly increased the collection of this information (88.4%) compared to a study of mystery clients reporting TB-like symptoms with the same population of providers in 2019 (25.7%) [2]. Their performance was similar to that reported in the public sector in 2022 (88%) [1], where TB-HIV integration has been a high priority. Anecdotally, GPs had indicated a reluctance to ask about HIV status for fear of offending clients. The intervention suggests that enquiring about HIV in this high-burden population can become routine practice for private-sector GPs with prompting. This practice is necessary for following appropriate TB testing protocols and managing clients. Under South African guidelines, negative TB tests by GeneXpert should be followed by TB culture testing among people living with HIV to ensure TB is not missed [3]. Ascertaining HIV status is essential for linkage to antiretroviral

treatment which is free in South Africa's public system, and TB preventive treatment, if appropriate [18].

## Implications

The intervention showed a workable public-private mix model for TB management in a high TB burden setting in South Africa. Through a collaboration with local government and the use of an existing medical referral application (Vula), private GPs were willing and able to screen clients for TB, initiate onsite sputum specimen collection, diagnose and refer or support clients with TB, and improve HIV enquiry, all without monetary incentive. Providing access to TB testing through the public sector reduced TB diagnostic delays and financial burdens for clients. Recently a National Health Insurance Bill was signed into South African law, mandating the increased integration of the public and private sectors, especially with respect to the management of TB and HIV [19]. Ensuring private providers can access GeneXpert Ultra tests through the public system will be an essential step to achieving this goal.

Our findings also suggest that, while concerning, the burden of TB in the private sector is substantially less than in the public sector. While the high yield reported in our study (17.6%) suggests underscreening, the total number of people diagnosed with TB (n=107) amongst 61 private GPs was far fewer than the burden in the district's public sector, which reported 14 538 positive GeneXpert Ultra tests (for all specimen types) among 17 public hospitals, eight community health centres, and 106 clinics over the same time period. The higher yield in the private sector suggests that enhanced screening is a priority for the sector. Yet, the lower TB burden suggests that linking private GPs to public GeneXpert Ultra testing is feasible in that it is unlikely to overtax the system. At present, private GeneXpert Ultra testing is both inconvenient, requiring a private laboratory visit, and cost prohibitive (US$26-70) for the client – a cost that is not covered by medical insurance for those fortunate enough to have it, unless the test result is positive [2,20]. Initiating sputum collection at the point of care reduced the burden on private clients, encouraged GPs to investigate TB at first encounter, and likely reduced delays in diagnosis and treatment initiation. As client-friendly services are a priority to the National TB Programme [21], key benefits of the intervention were the ability to reduce client travel and wait times while also reducing treatment delays.

Given these insights, we propose the following recommendations as a result of study findings:

- Action is needed to improve GP awareness of TB in the private sector and update GPs on national guidance for screening and testing. This could be achieved through government-endorsed webinars (e.g. through the South African knowledge hub), engagement with Independent Practice Associations, and through development of private practice policies for TB and HIV screening and testing under NHI.

- Private GPs should be encouraged to collect sputum specimens on-site and to enquire about HIV whenever TB is suspected to hasten appropriate testing and treatment. Our findings suggest these steps are acceptable and feasible to clients and GPs. What happens with sputum specimens will have to be determined by setting and resources. In some settings, transporting sputa to the nearest public clinic to be collected by laboratory services may be the most practical. In settings with a higher density of GPs, dedicated transport services to an NHLS laboratory may be more economical. In our study, dedicated drivers were employed to collect and deliver specimens. Outsourcing this service to transport companies and/or use of scooters for transport between clinics or to a central laboratory could improve the affordability of the model. An agreement with NHLS transport services

to collect specimens upon request as part of routine services might also be considered. Analysing cost effectiveness of various iterations of our model would be a good next step to see what would be most appropriate. We also recommend the use of the Vula application for coordinating specimen transport and prompting HIV enquiry, as Vula is already widely used in South Africa [22], and the study-developed form can be made freely available to GPs across the country. Further consideration should also be given to private providers' access to public stock of TB medications for uncomplicated TB, as has been proven beneficial elsewhere [23]. Some clients indicated a preference for continued private care; however, most changed their minds when confronted with an actual TB diagnosis and associated medical expenses, e.g. treatment and follow-up care. Many private providers are already registered as locations where people living with HIV can access antiretroviral therapy provided through the public system [24]. Expanding this model to include treatment for drug-susceptible TB is a priority for the national government [25], and uptake of our intervention suggests a willingness from the private sector to provide this support. Modest public incentives to encourage GPs to provide free follow-up care to clients diagnosed with TB may also be considered to improve continuity of care and relieve some strain on the public sector. Based on our experience and others in the private-public space, we further recommend the involvement of an intermediary such as non-governmental organisations or practice associations who can understand the differing needs of the private and public health environments [26].

## Limitations

Our study had several limitations. Firstly, we were unable to directly compare TB diagnoses and outcomes to previous private sector data. While there are mechanisms for private providers to notify the national TB programme of TB diagnosed within their practices, the government does not distinguish TB notifications that originated in the private sector.

It is also important to note that clients in the present study were provided with telephonic adherence support, which was not provided publicly or privately as the standard of care during the study period, thus client outcomes were likely to be better than those of clients with no adherence support. A study laboratory technician was also hired to process specimens collected privately. Had the identical specimens been submitted within the regular NHLS system, time to diagnosis and treatment initiation is likely to have been longer. Nonetheless, comparison with data from other South African studies in the private sector suggests that the collection of sputum specimens by private GPs and connection to public testing mechanisms likely reduced the time to diagnosis and treatment initiation considerably [2,14].

Another limitation is that it is not known how many participating GPs would have referred the 107 clients for TB testing in the public sector without their involvement in the study. However, previous research in the same GP population suggests that the intervention likely improved client outcomes by initiating TB testing at the first encounter [2].

Other limitations include study duration, overlap with the COVID-19 pandemic, and the potential for participation bias. The intervention was limited to 11 months, with increasing numbers of GPs signing on during the course of the study. As the practice of sputum collection and transport requests in Vula became normalised, GPs may have continued to increase their TB screening practices. Additionally, initiating the study during the height of the COVID-19 pandemic may have reduced the uptake and participation of some providers due to hesitancy to collect sputum onsite. If the intervention were to become standard practice, there would likely be increased uptake over time. On the other hand, 61 GPs declined to participate due to lack of interest and 21 indicated they do not

see clients with TB. GPs who chose to participate in the study may differ in systematic ways from those who did not. For instance those who are more interested in TB may have clients with higher risk factors and/or take different care approaches than those who did not participate. While we tried to minimise these biases by recruiting GPs from different practice areas, participation bias may limit the generalisability of our findings to the broader practice area.

There are a number of areas for future research that arose from this study. These include understanding the needs of GPs who declined to participate and barriers to sputum specimen submission from GPs practising in townships, where TB incidence is often higher. Assessing cost benefit and cost effectiveness would also be important for consideration of wider scale-up, factoring in costs of later stage presentations that may result from a lack of intervention. Future research might also consider the willingness of GPs to offer follow-up drug-susceptible TB services at a reduced rate – or free with modest government incentives – with access to publicly funded TB treatment.

## Conclusions

In general, the pilot was successful. Almost 20% of GPs in eThekwini health district collected and submitted sputum specimens without monetary incentives, a figure likely to be higher in the absence of COVID-19 pandemic conditions. The study identified and helped link 100 clients to TB treatment expeditiously. Most treatment occurred in the public sector (96.0%), although providing public stock of drug-susceptible TB treatment to private GPs would help to further unburden the public system. The high diagnostic yield reported (17.6%) suggests that more TB screening is needed in the private sector, while the low number of people diagnosed with TB compared to the public sector suggests that connecting private GPs to public TB testing is a sustainable option. Although the proportion of men to women diagnosed with TB mirrored that in the public sector, a higher proportion of men were tested in the private sector (1.1:1), suggesting that TB screening in this sector may be a helpful approach to identify more men at earlier stages of disease. Finally, user-prompting to check HIV status significantly improved the frequency with which GPs enquired about HIV to be more in line with the public sector.

## Acknowledgments

This study investigators are grateful for the support and collaboration of the KwaZulu-Natal Department of Health and the National Health Laboratory Service which provided GeneXpert Ultra testing and test kits. We would also like to thank the staff at eThekwini District Health Office who provided guidance on adherence support and assisted with clarification on national TB guidelines.

We would like to extend our thanks to the KwaZulu-Natal Doctors Healthcare Coalition and the South African Medical Association for raising awareness about the project. We would especially like to thank the GPs who took time to learn Vula and screen clients for TB and HIV, and the clients who participated in the study.

## Author contributions

**Conceptualization:** Jody Boffa, Tsholofelo Mhlaba, Jeremiah Chikovore, Sizulu Moyo.

**Data curation:** Jody Boffa, Buyisile Chibi, Keeren Lutchminarain, Sizulu Moyo.

**Formal analysis:** Jody Boffa, Buyisile Chibi.

**Funding acquisition:** Jody Boffa, Tsholofelo Mhlaba.

**Investigation:** Jody Boffa, Mergan Naidoo, Keeren Lutchminarain, Khine Swe Swe-Han, Jeremiah Chikovore, William Mapham, Sizulu Moyo.

**Methodology:** Jody Boffa, Tsholofelo Mhlaba, Mergan Naidoo, Keeren Lutchminarain, Khine Swe Swe-Han, Jeremiah Chikovore, Sizulu Moyo.

**Project administration:** Jody Boffa, Tsholofelo Mhlaba, Buyisile Chibi.

**Resources:** Keeren Lutchminarain.

**Supervision:** Jody Boffa, Tsholofelo Mhlaba, Buyisile Chibi, Keeren Lutchminarain, Jeremiah Chikovore, Sizulu Moyo.

**Validation:** Buyisile Chibi, Keeren Lutchminarain, William Mapham.

**Visualization:** Jody Boffa.

**Writing – original draft:** Jody Boffa.

**Writing – review & editing:** Tsholofelo Mhlaba, Buyisile Chibi, Mergan Naidoo, Keeren Lutchminarain, Khine Swe Swe-Han, Jeremiah Chikovore, William Mapham, Sizulu Moyo.

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
