## [Decision Letter · Decision Letter 0]

1 Nov 2024

PGPH-D-24-01929

Improving tuberculosis diagnosis in South Africa’s private sector: The results of a pilot public-private mix intervention in eThekwini health district

Dear Dr. Boffa,

Thank you for submitting your manuscript to PLOS Global Public Health. After careful consideration, we feel that it has merit but does not fully meet PLOS Global Public Health’s publication criteria as it currently stands. Therefore, we invite you to submit a revised version of the manuscript that addresses the points raised during the review process.

Please ensure that your decision is justified on PLOS Global Public Health’s publication criteria and not, for example, on novelty or perceived impact.

We look forward to receiving your revised manuscript.

Kind regards,

Giorgia Sulis, M.D., Ph.D.

Academic Editor

Journal Requirements:

 1. When completing the data availability statement of the submission form, you indicated that you will make your data available on acceptance. We strongly recommend all authors decide on a data sharing plan before acceptance, as the process can be lengthy and hold up publication timelines. Please note that, though access restrictions are acceptable now, your entire data will need to be made freely accessible if your manuscript is accepted for publication. This policy applies to all data except where public deposition would breach compliance with the protocol approved by your research ethics board. If you are unable to adhere to our open data policy, please kindly revise your statement to explain your reasoning and we will seek the editor's input on an exemption. Please be assured that, once you have provided your new statement, the assessment of your exemption will not hold up the peer review process. 2. Please provide separate figure files in .tif or .eps format. For more information about figure files please see our guidelines:  https://journals.plos.org/globalpublichealth/s/figures https://journals.plos.org/globalpublichealth/s/figures#loc-file-requirements  3. "Figure 3": please (a) provide a direct link to the base layer of the map (i.e., the country or region border shape) and ensure this is also included in the figure legend; and (b) provide a link to the terms of use / license information for the base layer image or shapefile. We cannot publish proprietary or copyrighted maps (e.g. Google Maps, Mapquest) and the terms of use for your map base layer must be compatible with our CC-BY 4.0 license.  Note: if you created the map in a software program like R or ArcGIS, please locate and indicate the source of the basemap shapefile onto which data has been plotted. If your map was obtained from a copyrighted source please amend the figure so that the base map used is from an openly available source. Alternatively, please provide explicit written permission from the copyright holder granting you the right to publish the material under our CC-BY 4.0 license. Please note that the following CC BY licenses are compatible with PLOS license: CC BY 4.0, CC BY 2.0 and CC BY 3.0, meanwhile such licenses as CC BY-ND 3.0 and others are not compatible due to additional restrictions.  If you are unsure whether you can use a map or not, please do reach out and we will be able to help you. The following websites are good examples of where you can source open access or public domain maps: * U.S. Geological Survey (USGS) - All maps are in the public domain. (http://www.usgs.gov) * PlaniGlobe - All maps are published under a Creative Commons license so please cite “PlaniGlobe, http://www.planiglobe.com, CC BY 2.0” in the image credit after the caption. (http://www.planiglobe.com/?lang=enl) * Natural Earth - All maps are public domain. (http://www.naturalearthdata.com/about/terms-of-use/)

Additional Editor Comments (if provided):

The manuscript is very interesting and addresses an important issue. However, our reviewers have raised some questions and concerns around the methodology and reporting of findings that require clarification. Please address these points in revision and provide a detailed explanation in a rebuttal letter.

Reviewers' comments:

Reviewer's Responses to Questions

**Comments to the Author**

1. Does this manuscript meet PLOS Global Public Health’s publication criteria? Is the manuscript technically sound, and do the data support the conclusions? The manuscript must describe methodologically and ethically rigorous research with conclusions that are appropriately drawn based on the data presented.

Reviewer #1: Yes

Reviewer #2: Yes

Reviewer #3: No

2. Has the statistical analysis been performed appropriately and rigorously?

Reviewer #1: Yes

Reviewer #2: Yes

Reviewer #3: No

3. Have the authors made all data underlying the findings in their manuscript fully available (please refer to the Data Availability Statement at the start of the manuscript PDF file)?

Reviewer #1: Yes

Reviewer #2: Yes

Reviewer #3: Yes

4. Is the manuscript presented in an intelligible fashion and written in standard English?

Reviewer #1: Yes

Reviewer #2: Yes

Reviewer #3: Yes

5. Review Comments to the Author

Reviewer #1: Well done to the authors on a very well written paper on an important topic. There is very limited published work on TB in the private sector in South Africa. So. this paper will add scientific knowledge to this area of study.

I only have 2 minor issues:

1. Table 1: Please check the table as there appear to be ad hoc numbers down the right hand side

2. Limitations: I think the authors could explore more fully how the study results may be biased by the GPs who opted not to be included.COVID could have played a role as the authors point out, but COVID would have affected all GPs. What other underlying issues could also apply? e.g. disinterest in TB/thoughts that it is a disease that should be dealt with in the public sector etc. It is interesting that a high % of GPs who opted out had practices in township areas.

Reviewer #2: This is an important contribution to TB diagnosis in the private sector. The private General Practitioners (GPs) however 96% opted for treatment in the public sector due to free access and follow up. The GPs thus were involved in the initial assessment and referral and not continued care and support for those with TB. Of note is the query on HIV status of the patients with TB symptoms which is a positive result owing to the information system prompting. Specimen yield and quality were high and perhaps a comment on why this was possible - ? better training of GPs and enhanced communication with patients; rapid turn around time to testing (dedicated transport and dedicated laboratory support). Initiation of treatment timeously must be commended but may be due to the previuosly described factors.

A useful benchmark will be the cost benefit and cost effectiveness of the use of the private sector (noting no incentives) but costs of training, costs of additional support of infomation system, dedicated transport for specimens, dedicated laboratory testing and adherence support.

The key issue of course is the involvement of 50.5% of GPs and submission of specimens by 19.5% of GPs. Overall 100 patients with TB were diagnosed and linked to care including some with drug-resistant TB who may have been missed in the public sector. Also of note is the heightened awareness of the 158 GPs and the 128 GPs that were trained as per study requirements.

Of course, it is not known how many GPs would have referred a patient with TB sysmptoms to the public sector without their involvement in the study.

Reviewer #3: The authors describe the evaluation of an intervention to improve the diagnosis of TB and HIV in private practices in a reaonably defined area in South Africa. This is a logical follow up to the referenced studies evaluating the effectiveness of private practioners in diagnosis of TB and HIV in sample patients (actors).

As the authors note, there is no defined population for the patients within the practices from whom 50% of invited physicians agreed to participate nor for the 39% who submitted patient sputum specimens during the study. There is no valid comparison for the number diagnosed nor the frequencly by individual physicians and the comparison with the national number and frequency is not valid.

The authors state that recruitment of physicians was ongoing during the study, but then a callendar-defined 11 month of data collection is described. Presumably some physician began the fiirst month and completed 11 months and others only a few months? If so this makes Table 1 impossible to intrepret. How many practice-months were in the study? Row percentages rather than columns would be of more interest.

The findings as presented are that some physicians who agreed to participate and accepted "training." When they were provided with an individual capable of promtly transporting the specimens to the public lab, they were able to diagnose TB and collect HIV diagnoses (only a few with a new HIV diagnosis?).

The important questions for this implementation study (reference #9 an unpublished abstract is not helpful) are:

1. What were the physicians asked to do for training and was the length of committment to "training" a barrier?

2. Were there other reasons for not participating?

3. Was compensation for training time provided to participants? Would that be required for a non-study implementation?

4. How was the transportation of specimens funded? If so, how would this be funded more broadly?

5. Do the authors conclude this approach warrant implementation or further study? What are the necessary components beyond providing rapid, accurate specimen testing and public health treatment access?

6. PLOS authors have the option to publish the peer review history of their article (what does this mean?). If published, this will include your full peer review and any attached files.

**Do you want your identity to be public for this peer review?** For information about this choice, including consent withdrawal, please see our Privacy Policy.

Reviewer #1: **Yes: **Sue-Ann Meehan

Reviewer #2: **Yes: **Barry Kistnasamy

Reviewer #3: **Yes: **Randall Rockne Reves

---

## [Editor Report · Decision Letter 1]

13 Jan 2025

Improving tuberculosis diagnosis in South Africa’s private sector: The results of a pilot public-private mix intervention in eThekwini health district

PGPH-D-24-01929R1

Dear Dr Boffa,

We are pleased to inform you that your manuscript 'Improving tuberculosis diagnosis in South Africa’s private sector: The results of a pilot public-private mix intervention in eThekwini health district' has been provisionally accepted for publication in PLOS Global Public Health.

Best regards,

Giorgia Sulis, M.D., Ph.D.

Academic Editor